# Point Cloud Registration via Heuristic Reward Reinforcement Learning

Bingren Chen

Data Mining Laboratory, Dalian University of Technology, Dalian 116000, China; bingren_chen@126.com

**Abstract:** This paper proposes a heuristic reward reinforcement learning framework for point cloud registration. As an essential step of many 3D computer vision tasks such as object recognition and 3D reconstruction, point cloud registration has been well studied in the existing literature. This paper contributes to the literature by addressing the limitations of embedding and reward functions in existing methods. An improved state-embedding module and a stochastic reward function are proposed. While the embedding module enriches the captured characteristics of states, the newly designed reward function follows a time-dependent searching strategy, which allows aggressive attempts at the beginning and tends to be conservative in the end. We assess our method based on two public datasets (ModelNet40 and ScanObjectNN) and real-world data. The results confirm the strength of the new method in reducing errors in object rotation and translation, leading to more precise point cloud registration.

**Keywords:** point cloud; registration; reinforcement learning; deep learning





## 1. Introduction

Point cloud registration is a primary task of high-quality 3D model reconstruction. A complete point cloud effectively captures the surface details of a detected object. However, due to the limitation of scanning equipment and environment, single-view point clouds with noise can inevitably be obtained. To get a complete point cloud of the object, multi-viewpoints need to be transformed into the same coordinate system, referring to the point cloud registration [1–3]. The iterative methods are applied in traditional algorithms, such as the Iterative Closest Point (ICP) algorithm. Despite being widely used, ICP is computationally expensive and demands the initial positions of two point clouds [4,5], which can sometimes lead to the local optimum.

In recent studies, some learning-based methods have been proposed to directly predict a transformation matrix for the source point cloud, e.g., ReAgent [6]. It handles point cloud registration using Imitation Learning (IL) and Reinforcement Learning (RL). An accurate initial policy can be obtained by imitating an expert, then fine-tuning the policy with a symmetry-invariant reward. ReAgent realized the registration step by step. This paper improves the model by addressing two limitations in ReAgent, including (1) the lack of the extraction of local features to point cloud in the state embedding stage, and (2) the fixed penalties for different states in the reward.

Specifically, with the purpose of getting the positional relation between the source and target point cloud accurately, and generating a more effective state representation, a feature extraction layer combined with EdgeConv is proposed, which enhances feature description in the state embedding. Furthermore, this paper defines a new reward function with time-varying penalties related to the current step. The new reward function allows more aggressive attempts at the early search stage while tending to be conservative over time. Finally, extensive experiments are conducted to assess the new method based on different public datasets and real-world data.

To sum up, the main contributions of this paper are threefold.

- First, to enrich the encoding process, an improved state-embedding module is proposed. It combines EdgeConv to capture the local features of relative coordinates, reflecting the key information of two point cloud positions among the current state.
- Second, a heuristic reward function is proposed. Unlike the invariable penalty in each step, the newly designed reward function allows aggressive attempts at the beginning when the environment is still unclear.
- Finally, the new method with an improved state-embedding module and the heuristic reward function is evaluated on two public datasets as well as real-world data of train components. The experimental results show that the new method effectively reduces the errors in rotation and translation, and can lead to more precise point cloud registration.

The rest of this paper is organized as follows. Related work is discussed in Section 2, and Section 3 introduces the three-dimensional point cloud registration methods in detail, including some principles of point cloud registration, an improved state-embedding module, and the stochastic reward function. Section 4 shows the experimental results to prove the effectiveness and feasibility of our methods. Finally, Section 5 concludes the paper.

## 2. Related Work

Point cloud registration methods can be generally divided into traditional algorithms and learning-based methods.

In traditional algorithms, coarse registration is usually the first step to making two point clouds closer. Some local feature descriptors need to be generated in point clouds [2,7–9], then similar features between the two point clouds should be identified. The key point matching algorithms are used to find the corresponding key point pairs, so that the correspondence between two point clouds is built up. The transformation matrix can be generated by the Singular Value Decomposition (SVD) method [10]. Even if some algorithms eliminate the mismatched corresponding key point pairs [11,12], there are still errors in the transformation, especially when two point clouds partially overlap. Therefore, the Iterative Closest Point algorithm (ICP) has been widely used as the fine calibration, as it minimizes the Euclidean distance between the point pairs, making the registration more precise. Although the requirements of ICP about the initial position and overlap rate of two point clouds are strict, the result of ICP still easily converges to the local optimum. Thus, improved work based on ICP was developed, aiming at solving problems such as the object in movement and the slow convergence rate [13–15]. The 3D-NDT method uses probability density to replace feature extraction and key point matching [3]. The 4-point congruent sets (4PCS) algorithm selects four points in the same base from the source point cloud and detects other four points in the same base from the target point cloud in the range of errors. Along this line, the correspondence is generated [16]. The traditional algorithms usually need to build up the correspondence or iterative calculation between the source and target point cloud.

On the other hand, learning-based registration methods make registration based on neural networks. As the pointwise network PointNet is proposed [17,18], other deep learning models are put forward gradually [19–21]. To extract the local features from point to point, the EdgeConv layer in DGCNN was proposed [22]. PointNetLK uses PointNet to extract features, then, the high-dimensional features are considered as the image to make the image registration so that the point cloud can be aligned [23]. Deep Closest Point (DCP) combines the feature-embedded network and an attention module to get a rigid transformation matrix [24]. Moreover, given the broad implementations of reinforcement learning (RL), e.g., classic games [25], quantitative trading [26], and image registration [27,28], RL has also been found useful in handling point cloud registration, and the ReAgent [6] that combined imitation learning (IL) and RL is a typical example. This paper focuses on improving the state embedding and reward function.

## 3. Methodology

This section introduces the heuristic reward reinforcement learning (RL) framework for point cloud registration. In general, an improved state-embedding module and the heuristic reward function are proposed to reduce transformation errors, improving registration more precisely.

### 3.1. Point Cloud Registration Network and ReAgent

Suppose there are two point clouds, the source point cloud $X$ and the target point cloud $Y$. The point set of $X$ may be the same as $Y$, indicating they are the same object. If $X$ is just a part of $Y$, then $X$ partially overlaps $Y$. There is a point cloud $X'$ in observation which need to be transformed to $X$, so the current observation can be defined as $O(X', Y)$, and the registration between $X'$ and $X$ can be written as follows:

$$X' \otimes T = X, \tag{1}$$

where $T$ is the transformation matrix. In traditional algorithms, based on the correspondence of two point clouds, the transformation matrix is usually obtained by SVD. However, the transformation matrix would be the prediction as the output by the learning-based methods.

The iterative registration is a kind of fine-tuning method with higher accuracy and inefficiency; it would transform the point cloud step by step instead of directly transforming in one shot. Suppose that the $X'$ requires $n$ steps to be aligned with $X$. That is:

$$X' \otimes T_1 \otimes T_2 \ldots \otimes T_n = X'_n = X \tag{2}$$

at step $i$, the point cloud $X'_i$ can be represented as:

$$X'_i = X'_{i-1} \otimes T_i \tag{3}$$

Thus, the transformation of the point cloud in each step can be regarded as the discrete actions in rotation and translation, and this process is similar to the learning process in RL. If the representation of state and reward function can be defined appropriately, RL techniques should be able to implemented.

Figure 1 shows the architecture of ReAgent for one iteration in step $i$. First, the features of $X'_i$ and $Y$ are embedded and concatenated to the state vector $S_i$ as the representation of the state, then it updates steps by using discrete, limited step sizes in each iteration. The policy $\pi(S)$ gives the probability of the actions that can be selected in the rotation and translation axis; it is computed by the action head of the agent. Additionally, it predicts the step sizes for this iteration. The disentanglement of rotation and translation is used to avoid errors in transformation.

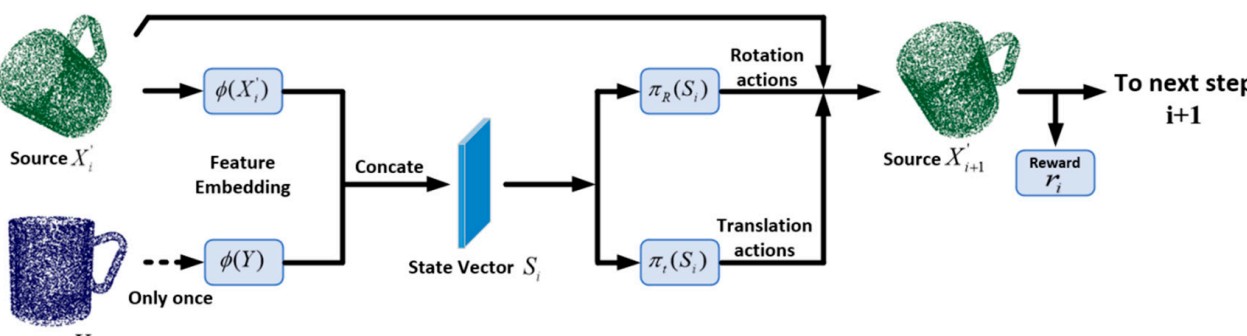

**Figure 1.** Architecture of ReAgent for one step.

Since point cloud registration is a complicated task, using RL to train the agent at the beginning may fall into the suboptimal policy, so the IL is used to train for the state embedding and the policy initialization.

### 3.2. Feature Extraction and an Improved State-Embedding Module

The representation of state is achieved by a PointNet-like architecture in ReAgent, and features of two point clouds are extracted separately with shared MLP layers.

In the process of state embedding, as shown in Figure 2, the target point cloud $Y$ is the extracted feature at the first step, then generated as $\phi(Y)$. The observed source point cloud $X'_i$ is the embedded feature at the beginning of each step. For example, $\phi(X'_i)$ represents the feature vector of the source point cloud in step $i$. Finally, $\phi(X'_i)$ is concatenated with $\phi(Y)$, and the state vector in step $i$ can be generated as follows:

$$S_i = concate\left[\phi(X'_i), \phi(Y)\right] \tag{4}$$

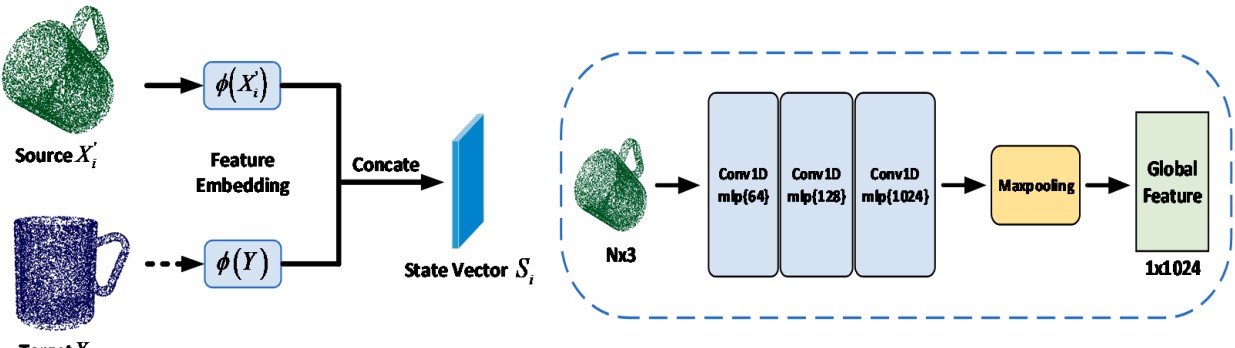

(**a**) State embedding        (**b**) The feature extraction layers

**Figure 2.** The architecture of feature embedding in ReAgent.

Specifically, in Figure 2b, the PointNet-like architecture is served as the feature embedding layer. The input is a point cloud with $N$ points and three-dimension information $xyz$. It increases the dimension of features through the one-dimensional convolution layer {64, 128, 1024}, then a $1 \times 1024$ global feature can be generated after max pooling, and two global features from $X'_i$ and $Y$ are concatenated to a $1 \times 2048$ state vector.

In ReAgent, fewer embedding layers are considered to learn the expressive feature vector sufficiently. However, some works like PointNet++ and DGCNN have proved that the local features are important to improve the accuracy of point cloud recognition and segmentation. DCP also uses DGCNN in feature extraction for better registration results. Although these local feature extraction methods require additional computing costs, local features about the positional relations between points and neighboring points are necessary for feature embedding. Based on extensive experiments, the new framework in this paper replaces the PointNet-like architecture by feature extraction layers combined with EdgeConv. The improved state-embedding module is shown in Figure 3a.

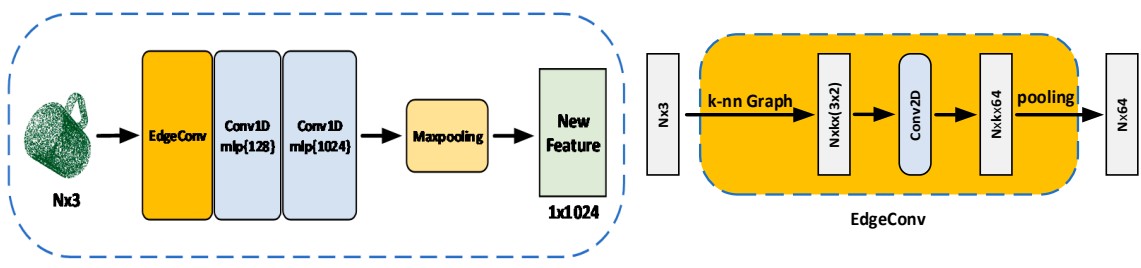

(**a**) Feature extraction combined with EdgeConv      (**b**) The EdgeConv architecture

**Figure 3.** Improved state-embedding module.

Furthermore, the input point cloud will go through an EdgeConv layer. Figure 3b shows that $k$ neighboring points can be detected by *k-nn* algorithm for each point in $N$,

computing the difference of three-dimension coordinate values between the *k* neighboring points and their center point; *k* vector point to the center point can be obtained, and a local neighborhood graph can be generated. After that, the local features between the center point and *k* neighboring points can be extracted by two-dimension convolutional layers. After pooling in dimension *k*, $N \times 64$ feature vector is obtained. The subsequent feature extraction architecture is similar to the ReAgent. Two $1 \times 1024$ feature vectors concatenate to a $1 \times 2048$ and serve as the state vector.

### 3.3. Heuristic Reward Function in RL

Only training the agent to imitate the expert policy does not guarantee consistently good performance in different datasets. Thus, ReAgent used RL to fine-tune and improve the model's generality. First, as an important evaluation measure, the Chamfer distance (CD) can be represented by:

$$CD(X,Y) = \frac{1}{|X|} \sum_{x \in X} \min_{y \in Y} \| x - y \|_2^2,$$ (5)

where $x \in X$ and $y \in Y$. Note that *CD* reflects the similarity of two point clouds in terms of the coordinate differences. In ReAgent, the reward function, denoted by *r*, is defined based on CD:

$$r = \begin{cases} -\varepsilon^-, CD\left(X_i', X\right) > CD\left(X_{i-1}', X\right) \\ -\varepsilon^0, CD\left(X_i', X\right) = CD\left(X_{i-1}', X\right) \\ \varepsilon^+, CD\left(X_i', X\right) < CD\left(X_{i-1}', X\right) \end{cases}$$ (6)

where *X* is the true source point cloud. It is identical to the target point cloud *Y*, or only partially overlaps *Y*. So, *X* is used to represent the point cloud that needs registration based on the observed point cloud $X^i$. Note that, $X_i'$ is a point cloud that is observed in step *i*, and $X_{i-1}'$ is the observed point cloud in step $i - 1$. The penalties $\left(\varepsilon^+, \varepsilon^0, \varepsilon^-\right)$ would be given depending on the *CD* at step *i* compared with the *CD* at step $i - 1$. The three penalties correspond to three reward states: "better", "same", and "worse", respectively. The values of $\left(\varepsilon^+, \varepsilon^0, \varepsilon^-\right)$ are set to (0.5, 0.1, 0.6) in [6].

If the CD between the current observed point cloud $X_i'$ and point cloud *X* is smaller than the last step, the transformation of point cloud $X^i$ in current step is considered as a "better" state, so the positive penalty $\varepsilon^+$ would be given. If the CD is larger or the same as last step, the transformation is considered as "worse" or "same". In these cases, the negative penalties $-\varepsilon^-$ and $-\varepsilon^0$ would be given respectively.

At the beginning of registration, there may be a large difference in rotation and distance between the point cloud $X^i$ and *X* at the initial position. In the process of iterative registration by RL, the actions selected by the agent's policy $\pi(S)$ in the first few steps may not reduce the values of *CD*, even making the *CD* increase.

Heuristic algorithms, such as simulated annealing that follow time-varying acceptance rates for new attempts, have been proven to be efficient in achieving global optimum. Examples of the implementations of heuristic methods in machine learning can be found in simulated annealing-based mobile sequential recommendation [29–31], stochastic deep learning [32], and stochastic subsampling RL [33,34].

Inspired by the simulated annealing algorithm, two parameters related to the current step are introduced to optimize the reward function:

$$\theta_m = t_m \cdot \alpha^i$$ (7)

$$\theta_n = t_n \cdot \beta^i$$ (8)

where *i* is the current step number, and $t_m, \alpha, t_n, \beta$ are set according to the experimental results. Then, the following heuristic reward function is proposed:

$$r = \begin{cases} -\varepsilon^{-} \cdot \theta_m, CD(X'_i, X) > CD(X'_{i-1}, X) \\ -\varepsilon^{0} \cdot \theta_n, CD(X'_i, X) = CD(X'_{i-1}, X) \\ \varepsilon^{+} \cdot \theta_m, CD(X'_i, X) < CD(X'_{i-1}, X) \end{cases} \tag{9}$$

where $\theta_m$ is a growing exponential function, and $\theta_n$ is a decreasing exponential function.

Therefore, in the first few steps, the actions selected by policy $\pi(S)$ may cause the values of CD to increase. Given the fact that the penalties of "worse" and "better" are small, while the penalty of "same" is relatively large, this reward function encourages the agent to take aggressive movements and avoid staying in the "same" state. In the last few steps, the penalties of "worse" and "better" states increase, leading to a more careful and accurate transformation by policy $\pi(S)$.

## 4. Experimental Results

This section discusses the results from experiments based on different datasets and robustness checks.

### 4.1. Registration on ModelNet40

First, I demonstrate the results from the experiments evaluating the new method, based on the ModelNet40 dataset. Following the same setting in [6], ModelNet40 has been split into two parts, the 1–20 categories models and the 21–40 categories. In the experiments, all models have taken resample, rigid rotation, and translation, so that the source and target point cloud can be obtained.

Based on imitation learning, the agent would be pre-trained for 50 epochs on the first 20 categories without any noise. Then, based on RL, another 50 epochs are supplied for fine-tuning the policy on the first 20 categories with some Gaussian noise added.

All experiments are performed under Windows11 operating system, Intel i9-12900k and 32 GB RAM, and RTX3090ti with the simulation software. We followed the parameters in [6]. The Proximal Policy Optimization (PPO) is used to update the policy, and the formulation in [35] can be implemented as used in actor-critic architecture. The PPO loss and advantage $\hat{A}$ are the same as the ReAgent. In the rotation and translation axis, there are 11 step sizes in each axis, [−0.27, −0.09, −0.03, −0.01, −0.0033, 0, 0.0033, 0.01, 0.03, 0.09, 0.27]. Note that the negative values indicate that the agent would take a transformation in negative directions of coordinate axes. The learning rate in pretraining by IL is set to 0.001 with halving it in each of the 10 epochs. The learning rate of RL is set to 0.0001. All the point cloud data would be pretreated according to the ReAgent.

We used several metrics that are commonly used in related work to evaluate performance.

Mean Absolute Error (MAE) is the error between the predicted vector $v_p$ and ground truth vector $v_{gt}$, and it can be calculated as following:

$$MAE_v = \frac{1}{3} \sum |v_p - v_{gt}| \tag{10}$$

where the vector can be a rotation or translation vector to calculate the errors.

Isotropic Error (ISO) only considers the values of rotation and translation matrix to calculate errors, so the ISO can be obtained as follows:

$$ISO_r = \arccos \frac{trace\left(R_d R_{gt}^{-1} - 1\right)}{2} \tag{11}$$

$$ISO_t = \parallel T_d - T_{gt} \parallel_2 \tag{12}$$

where *trace* is the sum of the diagonal elements of the matrix; $R_d$ and $T_d$ are the rotation and translation matrix in the end; $R_{gt}$ and $T_{gt}$ are ground truths for the point cloud to transformation.

The Chamfer Distance has been mentioned in the definition of the reward function. A Modified Chamfer Distance ($C\widetilde{D}$) is proposed by Lee and Yew [36]. It is defined as follows:

$$C\widetilde{D} = (P_s, P_t) = CD(P_s, P_{t,clean}) + CD(P_t, P_{s,clean}) \tag{13}$$

where *clean* means the point cloud with no noise.

Table 1 shows the experimental results on ModelNet40. Due to a different testing environment, the results of ReAgent were slightly different from the original paper, while the main patterns were found consistent. It can be seen that the new method and ReAgent obtained smaller errors of rotation and translation than the DCP-v2 [24] and PointNetLK [23]. Additionally, the errors of the new method are smaller than ReAgent in all 40 categories, while the running speed of ReAgent is faster. Although the running time of the new method is slower than ReAgent and DCP-v2, the accuracy of the new method in registration is better.

**Table 1.** Registration Results on ModelNet40.

| | The First 20 Categories | | | | | The Second 20 Categories | | | | | |
| | MAE | | ISO | | $C\widetilde{D}$ | MAE | | ISO | | $C\widetilde{D}$ | T |
| | R | T | R | T | ×0.001 | R | T | R | T | ×0.001 | (ms) |
|---|---|---|---|---|---|---|---|---|---|---|---|
| DCP-v2 | 3.876 | 0.032 | 7.826 | 0.071 | 2.81 | 4.912 | 0.038 | 9.138 | 0.079 | 3.95 | 21 |
| PointNetLK | 1.912 | 0.013 | 3.826 | 0.028 | 1.12 | 1.853 | 0.017 | 3.812 | 0.032 | 1.62 | 42 |
| ReAgent IL + RL | 1.783 | 0.011 | 3.189 | 0.024 | 0.76 | 1.760 | 0.011 | 2.996 | 0.023 | 0.99 | 19 |
| Our method IL + RL | 1.588 | 0.011 | 3.134 | 0.024 | 0.78 | 1.557 | 0.010 | 2.897 | 0.022 | 1.00 | 26 |

*4.2. Robustness Test*

To check the robustness of the model when noises exist, the different variance of Gaussian noise is respectively added to the point cloud, and the noise clipped to 0.05. Figure 4 shows the models with noise in different $\sigma$. The Chamfer Distances ($C\widetilde{D}$s) are calculated based on different results of registration.

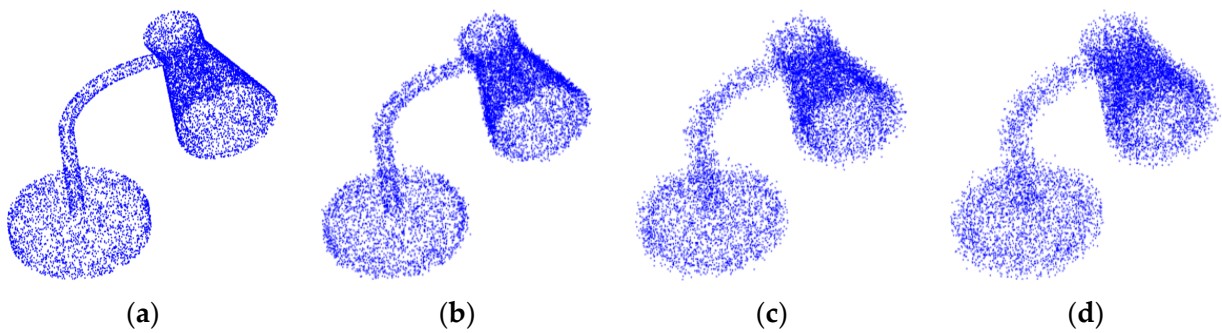

(**a**)　　(**b**)　　(**c**)　　(**d**)

**Figure 4.** Examples of point cloud with different variance of Gaussian noise. (**a**) Model without noise; (**b**) $\sigma = 0.01$; (**c**) $\sigma = 0.03$; (**d**) $\sigma = 0.05$.

The $C\widetilde{D}$-noise curve in Figure 5 shows the $C\widetilde{D}$ with different noise magnitudes. It can be seen that the values of $C\widetilde{D}$ obtained by the new method are consistently smaller than ReAgent, indicating that the values of the state embedding used some local features to represent the point cloud, and the heuristic reward function proposed in this paper. Overall, the results confirm the robustness of the new method under different noise levels.

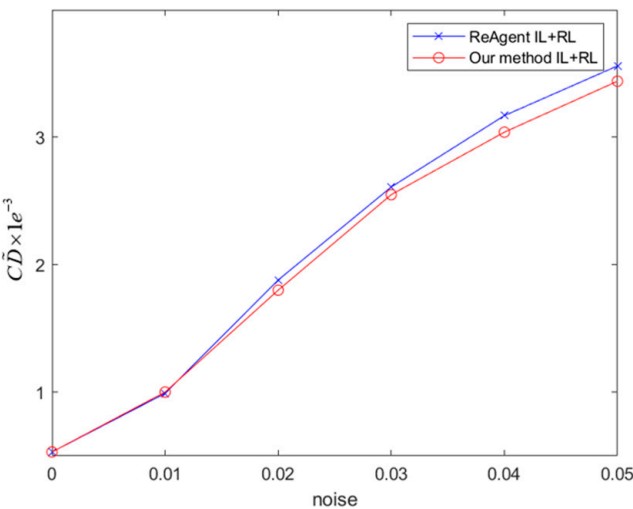

**Figure 5.** Performance Comparisons under Different Noise Levels.

### 4.3. Experiment on ScanobjectNN and Other Real-World Data

Experiments have also been conducted based on the ScanObjectNN dataset [37], which is collected from the depth sensor as the real data. The point clouds are segmented objects in ScanObjectNN, including 15 categories and 581 models in total, and 2048 points for each point cloud.

Furthermore, an additional category containing train components was manually collected (see [38]). Figure 6 shows 8 component point clouds such as traction rods, bolts, and wheelsets. They also have 2048 points with some noise after resampling. So, there are 16 categories and 589 point cloud models in the evaluated dataset.

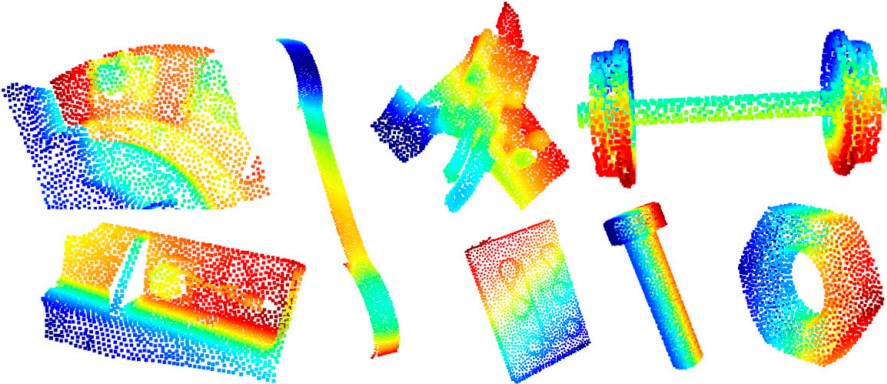

**Figure 6.** The train component point cloud.

Figure 7 shows the process of registration in train component wheelsets; the source point cloud (red) is transformed into the target point cloud (blue) step by step. The green point cloud represents the initial position of the source point cloud. Figure 7a demonstrates the results based on the ReAgent registration, and Figure 7b shows the results based on the newly proposed method. As can be seen, ReAgent results are unstable. It transforms after two point clouds overlapped at step 7 and the final step. Compared with ReAgent, the new method was designed to transform more and more conservatively over time, hence led to a stabilized overlapping result.

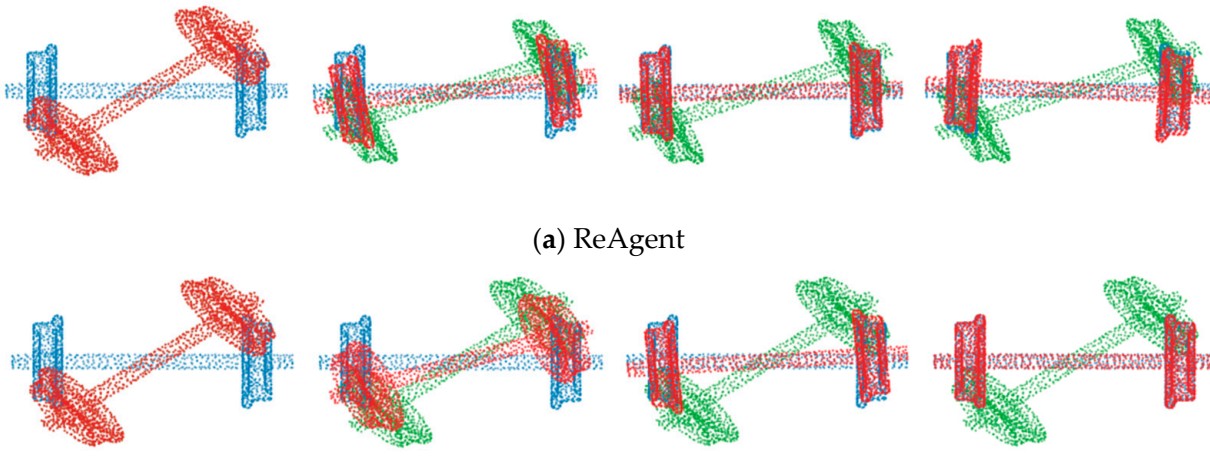

(**a**) ReAgent

(**b**) The proposed method

**Figure 7.** Example of component registration (initial position, step 1, step 7, and the final step).

As reported in Table 2, the errors of the new method are consistently smaller than DCP-v2, PointNetLK, and ReAgent. Importantly, the running time of the new method did not have significant changes compared with the running time in the ModelNet40-related experiments. It shows that the methods can be applied to practical applications on real-world data.

**Table 2.** Results on ScanObjectNN and train component.

|  | MAE | | ISO | | $\widetilde{CD}$ | T |
|---|---|---|---|---|---|---|
|  | R | T | R | T | ×**0.001** | ms |
| DCP-v2 | 8.760 | 0.081 | 17.320 | 0.163 | 5.08 | 53 |
| PointNetLK | 1.321 | 0.015 | 2.314 | 0.030 | 1.62 | 46 |
| ReAgent IL + RL | 1.449 | 0.012 | 2.789 | 0.025 | 0.75 | 22 |
| Our method IL + RL | 1.153 | 0.012 | 2.276 | 0.022 | 0.68 | 27 |

## 5. Discussion and Conclusions

Despite the overall outperformance of the new method which has been confirmed, there are two limitations that may lead to additional improvements in future work. First, since the EdgeConv has been used in embedding layers, the extraction of local features required an increasing computational complexity. Second, $\theta_m$ and $\theta_n$ related to the current step in the reward optimization were determined based on a series of experimental results. To address these two limitations, simplified but efficient embedding layers may be investigated so that the computing cost and the embedding effectiveness can be better balanced. Additionally, implementing the optimization process with parallel computing and high-performance computing techniques is also a possible research direction to enhance the computing efficiency while remaining the embedding quality. Furthermore, regarding the parameters, an adaptive time-dependent searching strategy may be developed for a more powerful optimization of the registration.

In conclusion, this paper introduces a point cloud registration method via heuristic reward reinforcement learning. An improved state-embedding module is also proposed to extract more local features about related positions from point to point. The heuristic reward function follows a time-dependent searching strategy, which allows aggressive attempts at the beginning and tends to be conservative in the end. The new method is evaluated on ModelNet40, ScanObjectNN, and additional real-world data, and the results confirm the improvements in terms of multiple evaluation metrics.

**Funding:** This research received no external funding.

**Institutional Review Board Statement:** Not applicable.

**Informed Consent Statement:** Not applicable.

**Data Availability Statement:** The ModelNet40 and ScanObjectNN are publicly available. Model-Net40 can be downloaded here: https://modelnet.cs.princeton.edu/, (accessed on 11 January 2023); The ScanObjectNN can be downloaded here: https://hkust-vgd.github.io/scanobjectnn/, (accessed on 11 January 2023).

**Conflicts of Interest:** The author declares no conflict of interest.

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
