# Peer review of "Point Cloud Registration via Heuristic Reward Reinforcement Learning"

_stats, doi:10.3390/stats6010016_

Round 1

Reviewer 1 Report

see comments in pdf file attached

Author Response

Summary of Revisions

Title: Point Cloud Registration via Heuristic Reward Reinforcement Learning

Authors: Bingren Chen

Manuscript ID: stats-2132777

Journal: Stats

I thank the editor and the anonymous reviewers for their valuable comments. They have helped to improve the paper in many ways. Important changes we made in the revised manuscript are summarized as follows.

  • First, I focused on improving the writing and clarity of this paper, all the phrases and sentences have been deliberated many times, and some vague expressions have been revised. The typos have been revised carefully.
  • Second, as per the suggestions from reviewers, I added more discussions in the paper, including the discussion about limitations and future work.
  • Finally, I added additional registration figures based on the same component data generated by the key baseline ReAgent. The comparison illustrates the strength of stability of the new method. Related discussions also have been added.

Response to Reviewer Comments

Responses to Reviewer 1

Point 1: Line 30: Spell out RL which appears later in line 93. Lines 78 & 88: Spell out the acronyms as done with most other terminologies - be consistent. Line 182: Replace the phrase “point distances” with “coordinate differences”. Line 238 & 239: correct typos to read 1e-3 and 1e-4, respectively.

Response 1: Thank you for the detailed comments. I have made changes in the revised manuscript following your suggestions. Also, I have carefully proofread the paper to improve the writing style.

Point 2: Line 183: Show or explain that the parameter r in equation (6) is related to the coordinate differences described in line 182.

Response 2: Thanks for your suggestion. Here  is the reward, which is defined by the coordinate differences (CD), as showed in equation (6). I have clarified them in the revision.

Point 3: Clarify the meaning on the six (6) values given in parentheses [0, 0.0033, 0.01, 0.03, 0.09, 0.27]. What do these values mean along the three positive coordinate axes (and in the negative directions)? Explain or show the function that defines the PPO.

Response 3: Thanks for your comment. The six values are the options of step size in each axis, represents the discretized actions on the three axes of rotation and translation. Adding the opposite direction of these positive moving steps, 11 values can be chosen, i.e., [-0.27, -0.09, -0.03, -0.01, -0.0033, 0, 0.0033, 0.01, 0.03, 0.09, 0.27]. Note that the negative values indicate that the agent can transform in negative directions of coordinate axes. According to the probability of selecting each step size, policy  selects the step size values and reflect them in corresponding axes. So does the policy  for the rotation, the values in the axes are computed by rotation axis and rotation angle, and the values represent the rotation component on each axis.

Point 4: How are these values are related the PPO, if at all? Explain or show the function that defines the PPO.

Response 4: As mentioned above, these values selected in axes of rotation and translation construct the action in the current iteration. After the transformation by action, the observed state would be the input for actor-critic architectures of PPO. I followed the process implemented in [38], and we clarified it in the revised manuscript.

Point 5: Figure 7. Figure shows point clouds of train components. This figure does not add materially to the thesis of this manuscript under review. Delete Figure 7 it is more distracting than beneficial.

The authors could use the space occupied by Figure 7 to include a new Figure that shows a step-by-step PC registration result from at least one of their real-world datasets. If this is not possible, then I recommend that the authors remove all references in the manuscript to the notion that the algorithm was tested on real-world datasets.

Response 5: Thanks for this comment. Figure 7 shows the initial position of the component wheelsets, step 1, step 7, and the final step, respectively. I agree that only showing the results from the new method was insufficient to demonstrate its strength. Hence, in the revised manuscript, we added additional sub-figures showing the results based on the ReAgent, the key baseline method. By comparison, I showed a better stabilization of the newly proposed method.

Reviewer 2 Report

1.   In the abstract, make sure to state the research contribution explicitly.

2.   What new IT knowledge is being created?

3.   In the paper, clearly state the problem's hypothesis.

4.   And reflect on the same conclusion in light of the facts identified.

5.   Discussion on the limitations of the work is missing.

6.   Please list the work's state-specific practical implications.

7.   Comprehensive editing of English language and style required:

A.     There are some sentence structure problems in the essay.

B.     Also, pay great attention to your spelling.

C.     For language correction needs, please use Grammarly.

Author Response

Summary of Revisions

Title: Point Cloud Registration via Heuristic Reward Reinforcement Learning

Authors: Bingren Chen

Manuscript ID: stats-2132777

Journal: Stats

I thank the editor and the anonymous reviewers for their valuable comments. They have helped to improve the paper in many ways. Important changes we made in the revised manuscript are summarized as follows.

  • First, I focused on improving the writing and clarity of this paper, all the phrases and sentences have been deliberated many times, and some vague expressions have been revised. The typos have been revised carefully.
  • Second, as per the suggestions from reviewers, I added more discussions in the paper, including the discussion about limitations and future work.
  • Finally, I added additional registration figures based on the same component data generated by the key baseline ReAgent. The comparison illustrates the strength of stability of the new method. Related discussions also have been added.

Response to Reviewer Comments

Responses to Reviewer 2

Point 1: In the abstract, make sure to state the research contribution explicitly.

Response 1: I appreciate your comment. To address the limitations of embedding and reward function in existing methods, an improved state-embedding module and a stochastic reward function are proposed. I revised the abstract to highlight these contributions.

Point 2: What new IT knowledge is being created?

Response 2: A heuristic-reward reinforcement learning framework for point cloud registration has been proposed. As a reduced number of embedding layers that extracted and concatenate the local features with global features, the description of states becomes more sufficient. Furthermore, a heuristic reward function follows a time-dependent searching strategy, which allows aggressive attempts at the beginning and tends to be conservative in the end.

Point 3: In the paper, clearly state the problem's hypothesis.

Response 3: The problem I solved in this paper is point cloud registration, which is a classic 3D task in computer vision. I also follow the testing environment of an existing study. Therefore, there is no newly proposed hypothesis. The model design and experiments focused on showing the improvement of the newly proposed method.

Point 4: And reflect on the same conclusion in light of the facts identified.

Response 4: The new method has been validated by following the same testing settings in an existing study. The experimental results showed that our method is more effective than the existing methods.

Point 5: Discussion on the limitations of the work is missing.

Response 5: Thanks for your comment. Following your suggestion, I have added a discussion of limitations and future work before the conclusion.

Limitation: Since the EdgeConv has been used in embedding layers, the extraction of local features can be computational expensive. Additionally,  and  which related to the current step are introduced to optimize the reward function, while the two parameters need to be determined based on experimental results. In the revised manuscript, I discussed these two limitations and provided possible solutions that may be done in the future work.

Point 6: Please list the work's state-specific practical implications.

Response 6: The point cloud registration is the key step of many 3D implications in detection and reconstruction, such as human faces recognition and reconstruction, tooth reconstruction.

Point 7: Comprehensive editing of English language and style required:

  1. There are some sentence structure problems in the essay.
  2. Also, pay great attention to your spelling.
  3. For language correction needs, please use Grammarly.

Response 7: I have carefully reviewed and proofread the paper several times to make sure that the grammar and other writing issues are minimized.

Reviewer 3 Report

the paper presents a Point Cloud Registration proposal via Heuristic Reward Reinforcement learning, it uses a stochastic reward function, the proposal is tested on 2 public data sets, here are some comments

·        minor grammatical corrections are recommended in the text, in general it is understandable and well written

·        It is recommended that the authors explain the way in which the learning of sub-optimal action policies is avoided with the proposal

·        One of the main problems of RL techniques is dimensionality, how is this problem avoided in the proposed method?

·        It is recommended to add in the conclusions the possible future work to extend the present method.

the paper is of scientific interest and shows an important contribution to the topic.

Author Response

Summary of Revisions

Title: Point Cloud Registration via Heuristic Reward Reinforcement Learning

Authors: Bingren Chen

Manuscript ID: stats-2132777

Journal: Stats

I thank the editor and the anonymous reviewers for their valuable comments. They have helped to improve the paper in many ways. Important changes we made in the revised manuscript are summarized as follows.

  • First, I focused on improving the writing and clarity of this paper, all the phrases and sentences have been deliberated many times, and some vague expressions have been revised. The typos have been revised carefully.
  • Second, as per the suggestions from reviewers, I added more discussions in the paper, including the discussion about limitations and future work.
  • Finally, I added additional registration figures based on the same component data generated by the key baseline ReAgent. The comparison illustrates the strength of stability of the new method. Related discussions also have been added.

Response to Reviewer Comments

Responses to Reviewer 3

Point 1: It is recommended that the authors explain the way in which the learning of sub-optimal action policies is avoided with the proposal.

Response 1: The new method reduces the initial penalties of “better” and “worse” through the parameters, but raise the penalties of “same”. Although it is not guaranteed that sub-optimal policies can be avoid, such design can solve some local optimal cases by encouraging aggressive attempts. This paper focuses on showing the effectiveness of RL in addressing the point cloud registration problem. In the future work, I should be able to improve the algorithm by considering stochastic search methods to address the potential sub-optimal issue.

Point 2: One of the main problems of RL techniques is dimensionality, how is this problem avoided in the proposed method?

Response 2: Thank you for the comment. Yes, dimensionality is an important issue that need to be considered in RL models. To address this problem, I follow the settings in ReAgent [6]. The initial transformation towards single point cloud is within a numeric range. The actions of rotation and translation are converted into discrete components of three axes.

Point 3: It is recommended to add in the conclusions the possible future work to extend the present method.

Response 3: Thanks for your comment. Following your suggestion, I have added additional discussion about limitations and future work before the conclusion.

Round 2

Reviewer 2 Report

I thank the author for the thoroughness of the reviewer's comments and suggestions.   They have contributed to the betterment of the manuscript in many respects, including important changes in the examiner's notes that you submitted with the revised manuscript.